# Genetic Landscape and Clinical Features of Hyperphenylalaninemia in North Ossetia-Alania: High Frequency of P281L and P211T Genetic Variants in the *PAH* Gene

**DOI:** 10.3390/ijms25094598

**Published:** 2024-04-23

**Authors:** Inna S. Tebieva, Polina V. Mishakova, Yulia V. Gabisova, Alana V. Khokhova, Tamara G. Kaloeva, Andrey V. Marakhonov, Olga A. Shchagina, Alexander V. Polyakov, Evgeny K. Ginter, Sergey I. Kutsev, Rena A. Zinchenko

**Affiliations:** 1North-Ossetian State Medical Academy, 362003 Vladikavkaz, Russia; tebinna@mail.ru (I.S.T.); kaloeva-tamara1991@rambler.ru (T.G.K.); 2Republican Children’s Clinical Hospital, 362003 Vladikavkaz, Russia; gabis32@mail.ru (Y.V.G.); alansio89@mail.ru (A.V.K.); 3Research Centre for Medical Genetics, 115522 Moscow, Russia; rabota.bio@gmail.com (P.V.M.); marakhonov@generesearch.ru (A.V.M.); schagina@med-gen.ru (O.A.S.); polyakov@med-gen.ru (A.V.P.); ekginter@mail.ru (E.K.G.); kutsev@mail.ru (S.I.K.)

**Keywords:** hyperphenylalaninemia, phenylketonuria, *PAH* gene, Republic of North Ossetia-Alania, Ossetian population, frequent mutations

## Abstract

This study, conducted in the Republic of North Ossetia-Alania (RNOA), aimed to explore the genetic landscape of hyperphenylalaninemia (HPA) and phenylketonuria (PKU) in the Ossetian population using data from newborn screening (NBS). Through comprehensive molecular genetic analysis of 29 patients with HPA from diverse ethnic backgrounds, two major genetic variants in the *PAH* gene, P281L and P211T, were identified, constituting 50% of all detected pathogenic alleles in Ossetian patients. Remarkably, these variants exhibited an exceptionally high frequency in the Ossetian population, surpassing global prevalence rates. This study unveiled a notable prevalence of mild forms of HPA (78%), underscoring the importance of genetic counseling for carriers of pathogenic variants in the *PAH* gene. Moreover, the findings emphasized the necessity for ongoing monitoring of patients with mild forms, as they may lack significant symptoms for diagnosis, potentially impacting offspring. Overall, this research offers valuable insights into the genetic landscape of HPA and PKU in the Ossetian population.

## 1. Introduction

Hyperphenylalaninemia (HPA) encompasses a group of clinically similar yet genetically diverse conditions characterized by autosomal recessive inheritance, all linked to a compromised metabolism of the amino acid phenylalanine (Phe). This spectrum includes classical phenylketonuria (PKU) caused by a deficiency in phenylalanine-4-hydroxylase (PAH), resulting from mutations in the *PAH* gene. Additionally, HPA involves impaired metabolism of tetrahydrobiopterin (BH4) associated with mutations in five distinct genes (*PTS*, *GCHI*, *QDPR*, *PCBD*, *SPR*) [1].

PKU was initially characterized in 1934 by the Norwegian physician A. Følling as a central nervous system disorder leading to mental retardation, identified by a distinctive “mouse” odor [2]. PKU is a prevalent inherited condition affecting individuals of diverse ethnic backgrounds, with varying birth prevalence rates. Globally, many countries conduct newborn screening (NBS) programs to promptly identify PKU patients. The timely initiation of a Phe-exclusion diet can prevent the manifestation of clinical symptoms and significantly enhance the patient’s quality of life. The nationwide NBS program for PKU in Russia commenced in 1997. Since 2023, the program has expanded to incorporate screening for a total of 36 congenital diseases, encompassing 34 inborn errors of metabolism [3].

The average incidence of classical PKU is approximately 1:7000–1:10,000 newborns [4]. Notably, Türkiye exhibits the highest frequencies (1:2600–1:4370), whereas Japan (1:80,500–1:143,000) and Finland (1:200,000) record the lowest rates [5,6,7,8]. In the Russian Federation, the incidence of PKU varies from 1:850 in Karachay-Cherkessia to 1:18,000 in the Republic of Tyva, with a national average of 1:7142 [9,10].

Clinical classification delineates classical PKU by a Phe concentration above 20 mg/dL, moderate PKU from 10 to 20 mg/dL, and mild hyperphenylalaninemia (mHPA) from 2 to 10 mg/dL [11,12]. Furthermore, it has been established that PKU and mild hyperphenylalaninemia (mHPA) without clinical manifestations are allelic variants of diseases associated with the *PAH* gene. These variants induce mild and severe phenotypes, with residual phenylalanine-4-hydroxylase activity emerging as a significant determinant of the metabolic phenotype in PKU [13]. The phenylalanine hydroxylase gene (*PAH*, MIM 612349) was identified in 1985 on the long arm of chromosome 12 within the q22-24.1 region [14]. Presently, the database PAHvdb (Available online: http://www.biopku.org/home/pah.asp, accessed on 6 October 2023) lists over 3300 mutations in the *PAH* gene.

The study of PAH functional activity reveals that the clinical variability in PKU may stem from various factors, including variable functional activity associated with different mutations in the *PAH* gene [11,15]. In compound heterozygotes, the severity of the disease’s clinical manifestation is thought to be determined by the less “severe” of the two *PAH* mutations, and the presence of a mutation with high residual activity of the PAH enzyme (in %) leads to a milder phenotype [15]. In this study, the term “mHPA” is used to emphasize the mild course of PKU, while “PKU” highlights classical PKU with severe clinical manifestations. “HPA” is employed to describe the condition overall.

*PAH* gene mutations exhibit variations among ethnic groups and specific regions, influencing the prevalence of pathogenic variants in individuals with mHPA and PKU across diverse populations globally, including in Russia [9,16,17]. Notably, the distribution of frequent pathogenic variants varies, reflecting the genetic diversity within populations.

In Russia, the predominant mutations associated with PKU are as follows: R408W (61.4%), P281L (4.95%), IVS10-11G>A (4.1%), R261Q (3.3%), IVS12+1G>A (2.3%), R252W (1.82%), and R158Q (1.65%) [18]. These findings underscore the significance of considering regional and ethnic-specific mutation profiles in the evaluation and diagnosis of individuals with *PAH*-related disorders. Many regions and ethnic groups in Russia have yet to be analyzed regarding the spectrum of mutations and prevalence.

The Republic of North Ossetia-Alania (RNOA), situated in the North Caucasus region of the Russian Federation, serves as the focal point for our study. Dominated by the Ossetian population (68.1% of the Republic’s population), this region boasts a rich cultural and historical tapestry. Other ethnic groups include Russians (18.9%), Ingush (3.8%), Kumyks (2.8%), Armenians (1.8%), Georgians (1.0%), and various smaller groups, each comprising less than 1% of the total population [19]. The Ossetians, also known as Alans, constitute an Eastern Iranian ethnic group, communicating in the Ossetian Eastern Iranian language, a member of the Indo-European language family. The ethnogenesis of the Ossetian people stems from the amalgamation of Alanian tribes, hailing originally as Sarmatians, with the indigenous Caucasian population. Beyond the confines of Ossetia, Ossetians are dispersed across Russia, Georgia, Turkey, and various other countries. Within the boundaries of the RNOA, distinct sub-ethnic groups such as the Irons, Kudars (representatives of South Ossetia), and Digorians coexist. The religious landscape among Ossetians predominantly comprises Christianity, with a minority adhering to Islam, notably within the Digorian sub-ethnic group, along with Assianism (Ossetian folk religion). Remarkably, it is the Ossetians alone who have preserved their Alanian language and identity amidst the currents of history and cultural evolution [20]. Our previous analysis of marriage ethnic assortativeness in the population of North Ossetia showed a high rate of intra-ethnic marriages within the population of Ossetians [21]. We could hypothesize that this might result in the accumulation of population-specific genetic load in the RNOA.

The objective of this study was to assess the efficacy of NBS for PKU and to analyze the epidemiological, genetic, and clinical characteristics of this condition in the Republic of North Ossetia-Alania (RNO-Alania).

## 2. Results

From 15 June 1997, NBS for PKU has been implemented in the RNO-Alania. In the period from 1997 to 2022, a total of 230,457 newborns were born in RNO-Alania, with 223,737 (97%) undergoing screening for PKU through NBS. Among them, 46 newborns with HPA were identified, exhibiting Phe levels above 2 mg/dL. Subsequent neonatal and retesting of Phe levels revealed 10 patients with classic and moderate PKU (Phe 10–30 mg/dL) and 36 patients with mHPA (Phe 2–10 mg/dL). It is worth noting that no patients with PKU were revealed in RNO-Alania outside the NBS.

The birth prevalence of HPA in RNO-Alania during this period was calculated to be 1:4864 newborns (95% confidence interval (CI): 1:3704–1:6667). Further breakdown indicated the following: classical PKU—1:22,374 newborns (95% CI: 1:12,195–1:47,619); mHPA—1:6216 (95% CI: 1:4545–1:9090). The NBS coverage was representative, reaching 97%.

A confirmatory DNA diagnosis for PKU was performed at the Research Centre for Medical Genetics (RCMG). Molecular genetic testing on 29 patients from 28 unrelated families, whose parents appeared interested in DNA diagnosis, revealed biallelic mutations in the *PAH* gene for 26 of them (96.5%). The 29 patients included 9 patients with PKU from eight unrelated families (one family has two brothers with PKU who were not strictly compliant with the diet therapy) as well as 20 patients with HPA. In one patient with HPA, no mutations in the *PAH* gene were detected, but mutations in the *PTS* gene (NM_000317.3:c.315-1G>A(;)c.370G>T) were found during subsequent studies.

A three-stage confirmatory DNA diagnosis in patients with PKU from RNO-Alania identified 18 genetic variants in the *PAH* gene (Table 1). These variants were categorized into “mild” and “severe” mutations based on literature data on the known residual activity of the PAH enzyme. Mutations with residual protein activity of less than 10% were classified as “severe”, while those with more than 10% residual activity were considered “mild”, with likely sensitivity to tetrahydrobiopterin drugs [22].

In the analyzed sample, the most frequent genetic variant was P281L, constituting 33.33% (allele count = 18 alleles, including 4 in the homozygous state). Among representatives of the Ossetian titular nation, this variant was found with a frequency of 42.11%. P281L belongs to the “severe” mutations class, characterized by a residual activity of the PAH enzyme of 2%.

The second most frequent genetic variant was P211T, accounting for 16.67% of the total sample and 18.42% of Ossetians. The allele count of this variant is nine alleles, including two in the homozygous state. P211T is characterized by high residual activity of the PAH enzyme (72%) and, in most cases, leads to a mild clinical presentation of the disease.

The mutation R408W, frequent in European and Russian populations, had an allele count of seven alleles, all in the heterozygous state. Its frequency amounted to 13% in the total sample and 7.9% in Ossetians. R408W belongs to the “severe” class with low residual activity of the PAH enzyme (2%).

Allele counts of other mutations, including S16* (mild mutation with a protein level of 6%), F331S (severe mutation with residual enzyme activity less than 10%), and V177M (PAH activity unknown), were each two alleles, with a frequency of 3.6% in the analyzed patient cohort (0%, 5.3%, and 5.3% in Ossetians, respectively). Each of the twelve different genetic variants (with varying residual activity of PAH) was found in the heterozygous state once.

The severity of clinical manifestations in patients with PKU is influenced by the patient’s genotype, specifically, the biallelic combination of genetic variants with different residual activity of mutant PAH. As presented in Table 1, the identified genetic variants can be categorized into different classes based on their residual PAH activity and associated severity. Severe mutations include P281L (PAH activity 2%), R408W (2%), M1R (2%), F331S (<10%), and S16* (6%). For five mutations (V177M, R169H, IVS10-14C>G, L83Wfs*9, ex3del), residual PAH activity is not known. Eight variants exhibit high residual PAH protein activity and can be classified as mild mutations based on available data (P211T, R261Q, Q419R, I306V, A300S, E390G, V230I, A403V; Table 1).

Table 2 shows the ethnic composition of the sample, patients’ genotypes in the *PAH* gene, Phe values, adherence to nutritional therapy, and the presence of a disabling factor in the form of intellectual impairment. Ethnically, the sample of patients with mutations in the *PAH* gene (27 people) was predominantly Ossetian (70.4%). Other ethnicities represented included Russians (7.4%), Turks (3.7%), Kazakhs (3.7%), and Kumyks (3.7%), as well as patients from interethnic families (11.1%), including Ossetian/Armenian, Ossetian/Kabardian, and Kumyk/Korean. The majority of marriages in families with PKU patients were monoethnic (88.9%).

In two patients of Ossetian origin, only a single nucleotide variant was found after three-stage molecular genetic testing (Table 2). We could not exclude the existence of the second affected allele in them since no search for deep intronic or regulatory variants was performed. 

All patients with two severe mutations of the *PAH* gene exhibited a classic or moderately severe form of PKU (six patients) with elevated Phe levels at NBS and retesting (13–41 mg/dL), and maximum Phe values at follow-up ranging from 13.9 to 42 mg/dL. Unfortunately, in two cases, the parents declined nutritional therapy, resulting in disabilities in their children. The remaining four compliant patients are undergoing treatment, demonstrating preserved intellectual development and age-appropriate psycho-physical development.

Among the twelve patients, their genotypes featured one mild mutation in combination with a severe mutation, and two patients had two mild biallelic mutations (P211T/R261Q and P211T/P211T). These patients showcased preserved, age-appropriate intellectual development, although most were non-compliant with nutritional therapy. Notably, in these cases, Phe levels at NBS, retesting, and the maximum Phe values at follow-up did not exceed 10 mg/dL.

The acquired data align with the concept of genotype–phenotype correlations in patients with HPA and PKU. The presence of a mild mutation in the genotype tends to alleviate the severity of the phenotype, which is a correlation supported by the findings of the current study.

Considering the specific spectrum and frequencies of mutations in the *PAH* gene within the Ossetian population, a population frequency estimate was conducted. Heterozygous carriage of nine genetic variants in the *PAH* gene, prevalent in the North Caucasus and the Russian Federation, was analyzed among 207 healthy unrelated individuals of Ossetian origin (414 chromosomes analyzed).

Among the Ossetian population, four carriers of the P281L variant and four carriers of the P211T variant were identified, resulting in a carrier frequency of these variants at 3.86% (95%CI: 1.68–7.47%). The R408W variant, common among Europeans and Russians, was not detected in the population sample, indicating its low frequency in the Ossetian population (<1.77%). Consequently, it can be inferred that approximately every 26th (1:26) ethnic Ossetian is a carrier of one of the two pathogenic variants of the *PAH* gene that are frequent in this population.

## 3. Discussion

The birth prevalence of PKU in the Republic of North Ossetia-Alania (RNO-Alania) was determined based on NBS data spanning from 1997 to 2022. The average birth prevalence of PKU was found to be 1:4864 newborns, a rate typical for many populations worldwide and in Russia. It is noteworthy that the prevalence of the classical form of PKU was relatively low, affecting 1:22,374 newborns. In the majority of cases (78.26% with a birth prevalence of 1:6216), mHPA was diagnosed, characterized by Phe levels not exceeding 10 mg/dL. Importantly, all patients with mHPA exhibited preserved age-appropriate psycho-physical development and were not disabled. Such patients typically do not require nutritional therapy and do not suffer from mental retardation. However, if they have two pathogenic variants of the *PAH* gene and marry heterozygous carriers of pathogenic mutations, there is a 50% risk of giving birth to a sick child and a 50% probability of a healthy carrier. In cases of pregnancy planning, it is essential to determine the carrier status of mutations in their partners, followed by prenatal DNA diagnosis of the fetal genotype. This is crucial because offspring with two severe mutations may have a more severe clinical presentation of PKU.

Confirmatory DNA diagnostics for PKU were conducted on 29 patients from 28 unrelated families, as per parental requests. Molecular genetic studies revealed that 62.8% of the entire sample, representing patients from diverse ethnic backgrounds (Ossetians—70.4%, Russians—7.4%, Turkish—3.7%, Kazakhs—3.7%, Kumyks—3.7%, and patients from interethnic families—11.1%), were carriers of the following three major genetic variants: c.842C>T (p.Pro281Leu; P281L)—33.33%, c.782G>A (p.Pro211Thr; P211T)—16.67%, and c.1222C>T (p.Arg408Trp; R408W)—13%.

The most prevalent genetic variant identified was the P281L missense mutation in exon 7, resulting in the substitution of proline by lysine and associated with low PAH activity (2%). In the studied population of patients with HPA in the RNO-Alania, the allele frequency of the P281L variant was found to be 33.2% (allele count = 18), with ethnic Ossetians exhibiting a higher frequency at 42.1% (allele count = 16). Notably, in two patients (allele count = four), the variant was observed in the homozygous state, accounting for 7.4% of all patients or 10.5% in the group of ethnic Ossetians.

The P281L mutation holds significance as a frequent mutation, ranking fourth to fifth in prevalence in Russia with a frequency of 3.5% [27]. However, this frequency is notably lower than what is observed in the Ossetian population, where it is comparable to the frequency noted in the geographically close neighboring region among the Georgian population, which stands at 33.7% [18]. Considering that Ossetians are Iranian-speaking people and Georgians belong to the Kartvelian language family, the similar pattern of P281L frequency suggests a possible explanation rooted in the mutation’s origin from the autochthonous population of the Caucasus, predating the formation of modern ethnic groups in the region.

The second most frequent genetic variant identified was the P211T missense mutation, involving the substitution of proline with tryptophan at position 211. This variant had an allele frequency of 16.6% for the entire sample and 18.4% for the Ossetian subgroup. The P211T mutation has been reported in Sicily and among patients in the Karachay-Cherkess Republic (family case). It is characterized by a residual enzyme activity of 72%, leading to a mild HPA phenotype [13,28].

These two genetic variants, P281L (33.33%) and P211T (16.67%), collectively account for 50% of all detected pathogenic alleles in the *PAH* gene within the sample of patients from the Republic of North Ossetia. In the subgroup of ethnic Ossetians, these variants constitute 60% (P281L-42.11%, P211T-18.42%). Notably, the frequency of the R408W mutation, prevalent in the European population, was 13% among all patients and 7.9% among ethnic Ossetians. The P281L variant with a lower frequency than in Ossetians is found in Iran (frequency 5–11%), Armenia (5.8%), and Russia (2.9%), but it is also frequent at 33.7% in the region of Georgia, geographically close to Ossetia, where Ossetians also live [29,30]. Interestingly, it is also prevalent (33.7%) in the region of Georgia, geographically close to Ossetia, where Ossetians reside. This genetic variant’s high frequency in both Ossetians and Georgians suggests a potential shared origin and a mechanism of “fixation.” The former South Ossetian Autonomous District of Georgia, now the Republic of South Ossetia, had 48,146 registered Ossetians in 2015 [https://www.refworld.org/pdfid/4c11f5452.pdf, accessed on 20 January 2024], with ongoing mestisation and migrations between northern and southern Ossetians and Georgians, supporting the possibility of a common origin for the P281L genetic variant in these populations [31].

North Ossetia appears to be a region with possibly the highest prevalence of the P281L variant in the world, potentially exceeding or being comparable to its prevalence in the Georgian region. The high frequency of the P211T variant (16.67%) is a distinctive feature of the Ossetian population and may also be among the highest in the world. Globally, the allele frequency of the P211T variant is 0.2% [4,32], while in the geographically close Karachay-Cherkess Republic, it is reported to be 1.3% [9].

The third most common genetic variant identified in the nucleotide sequence of the *PAH* gene is the R408W missense mutation in exon 12, leading to the substitution of arginine with tryptophan in position 408 [18]. In our dataset, the allele frequency was 13% among all patients and 7.9% among ethnic Ossetians. A notable proportion of alleles (Table 2) is associated with patients of Russian, Turkish, and Kazakh origin. This variant represents the majority allele for Russian patients (51.8%), while in the neighboring Ossetian region of Georgia, it occurs at a lower frequency of 3.2% [33]. The variation in frequencies could be explained by the greater connectivity of Ossetia with the European population. The frequency of the R408W mutation exhibits significant variability across different countries and ethnic groups, ranging from 5 to 80% of all mutant alleles [32].

The V177M and F331S variants were detected twice in the compound heterozygous state, with an allele frequency of 3.6% for the whole sample and 5.3% for ethnic Ossetians, as both variants were found exclusively among ethnic Ossetians in this patient sample. The F331S mutation has been reported in Slovakia in a single case and in a sample of patients from the Karachay-Cherkess Republic [9]. The V177M variant has been described in a compound heterozygous state in patients with PKU from the Galicia region of Spain [32] and from Georgia [33]. In a study of 3362 unrelated subjects in the Turkish population, 20 heterozygous carriers of this variant were found (allele frequency 0.6%) [23]. The S16* mutation was detected twice in the compound heterozygous state (3.5%). Both variants were found in ethnic Kumyk and Kumyk/Korean families. This variant has been described in the Russian population with an allele frequency of 0.4% [4].

Another 12 mutations of the *PAH* gene were detected in a compound heterozygous state, each occurring once. Of these, variants A300S, E390G, V230I, M1R, A403V, and L83Wfs*9 were found in ethnic Ossetians, variants R261Q and R169H in mixed families of Ossetians/Armenians, Ossetians/Kabardins, respectively, and variants Q419R, I306V, IVS10-14C>G, and ex3del in families of Kumyk/Koreans, Russians, and Kazakhs.

Based on the clinical presentation and analysis, we can infer the likely nature of mutations for five genetic variants with unknown residual PAH activity. Notably, two patients with genotypes R408W/ex3del and R408W/IVS10-14C>G had high Phe levels at NBS and retesting (19–29 mg/dL) and elevated maximum Phe values at follow-up (19–21 mg/dL). Based on this, we propose that these two variants—ex3del and IVS10-14C>G—exhibit low residual PAH protein activity, indicating severe mutations. This conclusion is further supported by the predicted molecular consequences of these genetic variations. Specifically, ex3del results in the deletion of exon 3 of the *PAH* gene (c.169-1_352+1del), leading to an out-of-frame deletion, formation of a preterm termination codon, and the absence of a functional protein. The IVS10-14C>G intronic variant is predicted by SpliceAI, SPiP, and MaxEntScan to disrupt the normal splicing pattern, likely leading to an aberrant mRNA product.

In contrast, genetic variants V177M and R169H could be regarded as mild mutations. This assumption is supported by low Phe values at NBS and retesting (3.1–4.5 mg/dL) and maximum Phe values at follow-up (2.3–5.9 mg/dL) as well as the literature data [34]. The effect of the L83Wfs*9 variant could not be conclusively determined solely from the obtained data. Predicted molecular consequences suggest that this variant may lead to the formation of a premature stop codon and subsequent activation of mRNA degradation through the nonsense-mediated decay (NMD) mechanism [35], indicating its potential classification as a severe mutation. However, it is noteworthy that this variant was identified in a patient with mHPA, where no second variant was detected. It is plausible that the unidentified second variant could be mild, preserving residual PAH activity at a sufficiently high level to mitigate the observed phenotype. Further investigation into the genetic background of patients with mHPA is warranted to elucidate the interplay among different variants and their impact on the clinical presentation of HPA.

The severity of the disease is influenced by the genotype, with mild genetic variants demonstrating a milder phenotype of HPA with a dominant effect [11,36,37]. This correlation between the *PAH* genotype and the phenotype of PKU and mHPA patients was observed, e.g., the P211T variant confers low blood levels of Phe. Mild genetic variants showed a tendency to produce a mild PKU phenotype with a dominant impact. These variants could have a particular influence on the maternal Phe level in blood during pregnancy [38]. Severe consequences of maternal PKU syndrome can include microcephaly, mental retardation, growth retardation, and congenital malformations (including congenital heart disease) [39]. Therefore, patients with mild forms should be monitored at follow-up by geneticists for genetic counseling and Phe-level management during pregnancy. However, in the Russian Federation, when patients reach 18 years of age, they are removed from pediatric dispensary observation and are observed in the therapeutic service only in the case of severe intellectual impairment. Maternal PKU syndrome is still a challenging problem in many countries.

Based on the carrier frequency data (1:26), the expected frequency of the disease associated with P281L and P211T variants in the Ossetian population was estimated. The cumulative allelic frequency of P281L and P211T was found to be 1.93% (95% CI: 0.84–3.77%). The expected disease incidence was calculated to be 1:2678 (95% CI: 1:704–1:14,172). These values, while not statistically different (*p*-value = 0.461) from the observed incidence of HPA in RNO-Alania (1:4864), are slightly higher than what was observed. We could hypothesize that the real incidence of HPA in RNO-Alania might be lower than expected, potentially because of the mild nature of the P211T mutation, and P211T/P211T homozygotes might have Phe level lower than 2 mg/dL (taking into account 72% PAH residual activity upon P211T variant), and thus might not have been identified in NBS. This hypothesis gains further support from the allele frequencies observed for the P281L and P211T variants within the patient cohort. Despite an equivalent carriage frequency of these two variants, their prevalence among patients differs significantly: 42.1% for P281L and 18.4% for P211T (*p*-value = 0.045). Notably, homozygosity for the P281L variant was detected in two patients, while no cases of homozygosity for the P211T variant were identified. This discrepancy underscores the potential influence of genotype on the phenotypic expression of HPA and warrants deeper investigation into the genotype–phenotype correlations in this context. It is worth noting that clinical variability is a known cause of the discrepancy between variant frequency and homozygous disease occurrence [40].

In one case, the diagnosis of BH4-deficient hyperphenylalaninemia, type A (OMIM: 261640), was confirmed. A previously described [41] pathogenic variant in exon 6 of the *PTS* gene (chr11:112233487G>T), leading to an amino acid substitution at position 124 of the protein (p.Val124Leu, NM_000317.2), was identified in the heterozygous state. Further investigation of the uncovered coding regions of the *PTS* gene by direct Sanger sequencing revealed another previously described pathogenic variant in intron 5 of the *PTS* gene (chr11:112233431G>A), leading to the formation of a splice site mutation (c.315-1G>A, NM_000317.2) in the heterozygous state [18]. The frequency of this identified nucleotide sequence variant in the gnomAD v. 2.1.1 control sample is 0.0004129%. The genotype of the patient with HPA is c.315-1G>A(;)c.370G>T. This patient is Avarian and is not ethnically Ossetian. The frequency of mutations in the *PTS* gene accounted for 3.4% of the entire sample of HPA patients (29 patients), with the remaining mutations found in the *PAH* gene.

## 4. Materials and Methods

This study included an analysis of the results of NBS for PKU spanning from the latter half of 1997 to December 2022. Additionally, a retrospective examination of medical records for patients diagnosed with PKU since the implementation of NBS for this condition was conducted for follow-up analysis of laboratory and clinical data.

The responsibility for organizing screening activities in the country was entrusted to the Medical Genetic Consultation (MGC) of the Republican Children’s Clinical Hospital (RCCH). The screening process comprised three stages. In the 1st stage, samples were collected in maternity hospitals. The NBS was conducted with the informed consent of the parents. Capillary blood was collected from the heel on the fourth day of life for term newborns and on the seventh day for premature infants. The blood was applied to Schleicher & Schuell 2992 porous filter paper. After drying, the samples were packed and delivered to the NBS laboratory. If the baby was born at home, the pediatrician collected the sample for the NBS during a visit to the newborn. The 2nd stage represents biochemical marker analysis. The levels of biochemical markers in whole blood samples were determined using time-resolved immunofluorescence. DELFIA Neonatal (Perkin Elmer, Turku, Finland) and FAVR (RICO-Med LLC., Moscow, Russia) reagents were utilized on Wallac (Perkin Elmer, Turku, Finland) equipment. Fluorescence levels were measured on a Victor2 device. In the case of a deviation from the norm, the level from the primary blood sample was redetermined in duplicates. At the third stage of NBS, the concentration of Phe in a novel blood sample was assessed.

Following the third stage of NBS, the prescription of dietary therapy and regular medical follow-up for PKU patients was initiated at the MGC of the RCCH in the Republic of North Ossetia-Alania. Phe level monitoring was conducted at intervals of 3 to 6 months, employing the same methodology utilized during NBS. During these monitoring sessions, the maximum Phe value observed over the entire follow-up duration was recorded for each patient. However, precise age data at the time of recording were not documented.

DNA diagnosis was conducted at the Laboratory of Molecular Genetic Diagnostics 1 of the Research Centre for Medical Genetics (RCMG) during the period from 2017 to 2022. The study employed a series of molecular diagnostic methods performed in three consecutive stages. The 1st stage involved analysis of 25 frequent genetic variants of the *PAH* gene, including S16*, L48S, IVS2+5G>A, IVS2+5G>C, R111*, IVS4+5G>T, ex5del4154ins268, R158Q, D222*, R243Q, R243*, R252W, R261Q, R261*, E280K, P281L, A300S, I306V, S349P, IVS10-11G>A, E390G, A403V, R408Wp, Y414C, and IVS12+1G>A. The 2nd stage consisted of the analysis of the coding sequence of the *PAH* gene as well as ±20 intronic regions flanking exons by direct Sanger sequencing. At the 3rd stage, copy number variations in the *PAH* gene locus were investigated by Multiplex ligation-dependent probe amplification (MLPA) using the P055-D1 PAH MLPA mix (MRC-Holland, Amsterdam, The Netherlands) according to the manufacturer’s recommendations. If the diagnosis was not confirmed in the first three steps, the study proceeded to search for rare variants by direct automatic Sanger sequencing of the *PTS*, *QDPR*, and *GCH1* genes.

Genetic variants were named according to legacy nomenclature (based on biopku.org resource). Table 2 contains the HGVS designation of variants according to the NM_000277.3 transcript variant of the *PAH* gene. In cases of gross deletions or genetic variants in other than *PAH* genes, coordinates were referred to according to the hg38 human genome assembly. *PTS* gene single nucleotide variants are named based on the NM_000317.3 transcript variant.

The frequency of PKU per 1000 people was calculated by determining the ratio of patients identified through NBS to the total number of examined newborns in the RNO-Alania. The formula used was *p* = (*p1*/*n*) × 1000, where *p* is the PKU frequency per 1000 people; *p1* is the number of children diagnosed with classic (PKU) and mild (mHPA) forms of PKU; and *n* is the total number of observations with clinically different forms.

The heterozygous carriage of frequent genetic variants for the Russian Federation and the North Caucasus region (R408W (p.Arg408Trp), P281L (p.Pro281Leu), P211T (p.Pro211Thr), R261Q (p.Arg261Gln), R158Q (p.Arg158Gln), IVS4+5G>T, IVS10-11G>A, IVS12+1G>A, and Ex5del4154ins268) was analyzed in 207 healthy unrelated individuals of Ossetian ancestry up to the third generation. The calculation of heterozygous carriage of mutations in the *PAH* gene was performed using the formula 2*pq* = *m*/*N*, where 2*pq* is the proportion of heterozygotes (based on the Hardy–Weinberg equation); *N* is the size of the studied sample; and *m* is the number of identified mutation carriers. Based on this equation, *q* (alternative allele frequency) is calculated and used for the estimation of disease prevalence.

Statistical analysis was conducted using WinPepi v. 11.65 [42] (School of Public Health and Community Medicine, Hebrew University, Jerusalem, Israel).

Participants in this study provided written informed consent for voluntary participation, including the sampling of biological materials and the publication of data in the open press. For minors, consent was obtained from their parents. This study received approval from the ethics committee of the RCMG (Protocol No. 5 dated 20 December 2010).

## 5. Conclusions

In conclusion, this study provides valuable insights into the clinical and genetic characteristics of HPA in the Republic of North Ossetia-Alania. The analysis revealed a high frequency of specific genetic variants, particularly the P281L and P211T mutations in the PAH gene, among HPA patients in this region. These variants are associated with different levels of residual phenylalanine hydroxylase (PAH) enzyme activity, which correlates with the severity of the disease phenotype.

This study highlights the importance of genotype–phenotype correlations in HPA patients, demonstrating that the presence of mild mutations often mitigates the severity of the clinical phenotype. Furthermore, the investigation of carrier frequencies among healthy individuals provides valuable epidemiological data, aiding in estimating disease prevalence and informing genetic counseling practices.

Overall, this research contributes to our understanding of HPA in the Ossetian population and lays the groundwork for future studies aimed at improving diagnostic approaches and therapeutic outcomes in affected individuals.

## Figures and Tables

**Table 1 ijms-25-04598-t001:** Allele frequencies of the *PAH* gene mutations in patients from RNO-Alania.

N	Genetic Variant				Total Sample		Ossetians		Mutation Type	Residual PAH Activity (%) *
	Legacy Name	Position in cDNA (According to NM_000277.3) (or hg38 Genomic Coordinates in Case of Gross Deletion)	Predicted Change at Protein Level (According to NP_000268.1)	rsID †	Allele Count	Allele Frequency %	Allele Count	Allele Frequency %		
1	P281L	c.842C>T	p.(Pro281Leu)	rs199475627	18	33.3%	16	42.1%	Missense	0
2	P211T	c.631C>A	p.(Pro211Thr)	rs62514931	9	16.7%	7	18.4%	Missense	72
3	R408W	c.1222C>T	p.(Arg408Trp)	rs5030858	7	13.0%	3	7.9%	Missense	0–5
4	S16 *	c.47_48delCT	p.(Ser16Ter)	rs62642906	2	3.7%	-		Small frame-shifting deletion	6
5	F331S	c.992T>C	p.(Phe331Ser)	rs199475614	2	3.7%	2	5.3%	Missense	<10
6	V177M	c.529G>A	p.(Val177Met)	rs199475602	2	3.7%	2	5.3%	Missense	**
7	R261Q	c.782G>A	p.(Arg261Gln)	rs5030849	1	1.9%	-		Missense	23
8	Q419R	c.1256A>G	p.(Gln419Arg)	rs752255985	1	1.9%	-		Missense	70
9	I306V	c.916A>G	p.(Ile306Val)	rs62642934	1	1.9%	-		Missense	25
10	A300S	c.898G>T	p.(Ala300Ser)	rs5030853	1	1.9%	1	2.6%	Missense	65
11	ex3del	chr12:(?_102894655)_(102895185_?)del	p.?	n/i	1	1.9%	-		Gross deletion	**
12	E390G	c.1169A>G	p.(Glu390Gly)	rs5030856	1	1.9%	1	2.6%	Missense	54–85
13	V230I	c.688G>A	p.(Val230Ile)	rs62516152	1	1.9%	1	2.6%	Missense	52–63
14	R169H	c.506G>A	p.(Arg169His)	rs199475679	1	1.9%			Missense	**
15	M1R	c.2T>G	p.(Met1?)	rs62508575	1	1.9%	1	2.6%	Missense	2
16	A403V	c.1208C>T	p.(Ala403Val)	rs5030857	1	1.9%	1	2.6%	Missense	32–100
17	IVS10-14C>G	c.1066-14C>G	p.?	rs62507334	1	1.9%			Intronic variant	**
18	L83Wfs*9	c.248del	p.(Leu83TrpfsTer9)	n/i	1	1.9%	1	2.6%	Small frame-shifting deletion	**
19	n/i	n/i	n/i	n/i	2	3.7%	2	5.3%		
In total:					54	100%	38	100%		

**Note**: n/i—not identified; *—data on the enzymatic activity of genetic variants relative to the wild type from the PAHvdb database [www.biopku.org, assessed on 6 October 2023] and from literature sources [13,22,23,24,25,26]; **—activity not known, †—dbSNP accession number (rsID).

**Table 2 ijms-25-04598-t002:** Genotypes of patients in the *PAH* gene, Phe values in follow-up, compliance with diet therapy, and presence of a disabling factor in the form of intellectual impairment in patients of different ethnic groups in RNO-Alania.

The Ethnic Background of the Parents	Sex	Genotype in the *PAH* Gene	Phe (mg/dL) Neonatal	Phe (mg/dL) Retesting	max Phe (mg/dL)	Diet: Compliance/ Non-Compliance	IIm/D
**Ossetians**							
Ossetians	F	P281L †/P281L †	24	31	39	refusal to diet	+/+
Ossetians	F	P281L †/P281L †	7.3	14	17	compliance (Afenilak)	-/-
Ossetians	F	P281L †/A300S	2.3	3	5.7	non-compliance	-/-
Ossetians	F	P281L †/A403V	4.8	4.5	7.9	non-compliance	-/-
Ossetians	F	P281L †/E390G	5.5	5.5	5.8	non-compliance	-/-
Ossetians	F	P281L †/F331S †	6.3	5.9	13.9	compliance (Afenilak)	-/-
Ossetians	M	P281 †L/M1R †	14	41	42	compliance (Afenilak)	-/-
Ossetians	F	P281L †/V230I	3.2	3.2	5.9	non-compliance	-/-
Ossetians	F	P281L †/P211T	4.1	5.2	8.6	non-compliance	-/-
Ossetians	F	P281L †/P211T	3.8	3.5	6.5	non-compliance	-/-
Ossetians	M	P281L †/P211T	4.1	4.2	9.6	non-compliance	-/-
Ossetians	F	P281L †/P211T	9.5	4.6	5.56	non-compliance	-/-
Ossetians	M	P281L †/P211T	4.8	4.5	5.7	non-compliance	-/-
Ossetians	F	P211T/P211T	3.6	3.5	6.2	compliance	-/-
Ossetians	M	R408W †/V177M ^	2.3	3.1	2.3	non-compliance	-/-
Ossetians	M	R408W †/V177M ^	3.8	3.7	3.05	non-compliance	-/-
Ossetians	M	R408W †/F331S †	6.7	13	16.8	compliance up to 1 year (Afenilak)	-/-
Ossetians	F	P281L †/#	3	2.4	4.12	non-compliance	-/-
Ossetians	M	L83Wfs*9 ^/#	4.3	4.5	5.9	non-compliance	-/-
**Other ethnic groups**							
Russians	F	R408W/I306V	3.5	3.9	4.6	non-compliance	-/-
Russians	F	R408W †/ex3del ^	7.5	19	21	compliance (Nutrition and PKU Energy Concentrate)	-/+
Kazakhs	F	R408W †/IVS10-14C>G ^	19	29	19	compliance (Nutrition and PKU Energy Concentrate)	-/+
Turkish ‡	F	P281L †/R408W †	19	29	19	refusal to diet	+/+
Turkish ‡	M	P281L †/R408W †	19	29	19	not strictly compliance with diet	+/+
Kumyks	M	S16* †/P211T	3.4	4.5	5.2	non-compliance	-/-
**Patients from interethnic marriages**							
Ossetian/Kabardian	F	P281L †/R169H ^	2.7	4.3	4.7	non-compliance	-/-
Ossetian/Armenian	F	P211T/R261Q	4.3	5.9	10	compliance (Lofenalak)	-/-
Kumyk/Korean	F	S16* †/Q419R	5.5	6.1	4	non-compliance	-/-

**Note**: ‡—two siblings with PKU from one family; #—Patients in whom the second pathogenic variant could not be identified; IIm/D—Intellectual impairment/disability; †—severe mutations with low residual activity PAH; ^—residual activity of PAH protein is not known.

## Data Availability

The datasets used and/or analyzed during the current study are available from the corresponding author upon reasonable request.

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
