# Peer review of "Genetic Landscape and Clinical Features of Hyperphenylalaninemia in North Ossetia-Alania: High Frequency of P281L and P211T Genetic Variants in the PAH Gene"

_ijms, 2024, doi:10.3390/ijms25094598_

Round 1

Reviewer 1 Report (New Reviewer)

Comments and Suggestions for Authors

The manuscript is informative and provides new data on PKU frequency in a unique , special population. The results of the study may be useful for healthcare decisions not only in the Republic of  North Ossetia , but in other countries where Ossetian population is living ( Georgia, Turkey, etc.) . Some remarks for the authors : 1,/ The text on page 2, lines 77 - 79. are written in Russian language , not in English. 2./ After the Introduction not  'Results ' should be the 2. chapter, but Materials and Methods (page 3, line 101.) , now it starts on page 9, line 375, as Chapter 4. 3./ Is the Newborn Screening Program obligatory in North Ossetia? What is the practical management of the screening  if the child is born at home, and not in a hospital? How many inborn errors of metabolism are involved in the NBS ? These data seem to be important to compare the screening system with other countries. 4./ NBS data were collected in the time period of 1997 - 2022 (page 9, lines 376-377), and DNA diagnosis was performed in the years 2017 - 2022 ( page 9, lines 393-394.) The authors write : "Following confirmatory DNA diagnosis the prescription of dietary therapy and regular medical follow - up for PKU patients was initiated... "(page 9, lines 415 - 417.). The question is, what happened with the PKU patients who were scrrened by NBS but without DNA diagnosis, between 1997 and 2017? The aim of NBS is the fast, immediate start of dietary treatment . A clear explanation is needed. 5./ The form of journal names is not uiform in References , some ones are not in shortend but full name form (eg. No. 7 and 35 ).

The manuscript is recommended for publication after minor corrections.

Author Response

First of all, we would like to thank the Reviewers for their in-depth and detailed analysis of our manuscript. We have tried to fulfill all their remarks and comments and believe the manuscript to be significantly improved.

Reviewer 1

The manuscript is informative and provides new data on PKU frequency in a unique , special population. The results of the study may be useful for healthcare decisions not only in the Republic of  North Ossetia , but in other countries where Ossetian population is living ( Georgia, Turkey, etc.) . Some remarks for the authors :

1,/ The text on page 2, lines 77 - 79. are written in Russian language , not in English.

Reply: Thank you for your thorough review. We have addressed the issue by replacing the sentence with an accurate one.

2./ After the Introduction not  'Results ' should be the 2. chapter, but Materials and Methods (page 3, line 101.) , now it starts on page 9, line 375, as Chapter 4.

Reply: We have ensured compliance with the formatting requirements of the International Journal of Molecular Sciences for the preparation of the manuscript. As per these guidelines, the Materials and Methods section should be positioned after the Discussion section, and we have adjusted the manuscript accordingly.

3./ Is the Newborn Screening Program obligatory in North Ossetia? What is the practical management of the screening  if the child is born at home, and not in a hospital? How many inborn errors of metabolism are involved in the NBS ? These data seem to be important to compare the screening system with other countries.

Reply: We have revised and updated the details regarding the Newborn Screening (NBS) program in Russia, as outlined in lines 44-47. We have also included detailed information on obtaining informed consent and procedures for home deliveries in the Materials and Methods section (lines 385-392).

4./ NBS data were collected in the time period of 1997 - 2022 (page 9, lines 376-377), and DNA diagnosis was performed in the years 2017 - 2022 ( page 9, lines 393-394.) The authors write : "Following confirmatory DNA diagnosis the prescription of dietary therapy and regular medical follow - up for PKU patients was initiated... "(page 9, lines 415 - 417.). The question is, what happened with the PKU patients who were scrrened by NBS but without DNA diagnosis, between 1997 and 2017? The aim of NBS is the fast, immediate start of dietary treatment . A clear explanation is needed.

Reply: Thank you for bringing this to our attention. We have rectified the description of treatment initiation in the Materials and Methods section (lines 399-405). Your assistance is greatly appreciated.

5./ The form of journal names is not uiform in References , some ones are not in shortend but full name form (eg. No. 7 and 35 ).

Reply: We have corrected the references accordingly.

The manuscript is recommended for publication after minor corrections.

Reviewer 2 Report (New Reviewer)

Comments and Suggestions for Authors

The research describes the results of screening for two diseases, hyperphenylalaninemia (HPA) and phenylketonuria (PKU), in newborns from North Ossetia-Alania in a 25-year period. In the samples of newborns with positive screening results (immunofluorescence and measurement of phenylalanine concentration) whose parents agreed to further testing (n=29), the most frequent genetic variants in the PAH gene were searched using molecular genetic methods, and in case these previously described variants were not found, Sanger sequencing was performed. In total 28 patients, a combination of genotyping and sequencing revealed 18 previously noted variants in the PAH gene, while in one person there were found no changes in the PAH gene.

It seems to me that the genetic research on the blood samples of newborns was carried out methodologically correctly, however the manuscript has many shortcomings and inconsistencies. First of all, most of the manuscript is not written in the spirit of the English language, so it is difficult to follow. Not only that, but also often unexpected new moments occur.

I`ll show an example: the authors stated that the research was conducted on the population of the North Ossetia-Alania, which is in the Introduction described as an “Eastern Iranian ethnic group, communicating in the Ossetian Eastern Iranian language”. Then, in Table 2 in the Results section, ethnic background of parents of the PKU-affected children was written, and it was shown that although the most of the parents were of Ossetian origin, there were also some other ethnic groups. And finally, in the Materials and Methods (page 9, lines 399-401), after specifying which genetic variants in the PAH gene were analyzed, the authors wrote sentence that has no meaning and refers to allele frequency in Russia.

“1st stage includes analysis of 25 frequent genetic variants of the PAH gene included S16*, L48S, IVS2+5G>A, IVS2+5G>C, R111*, IVS4+5G>T, ex5del4154ins268, R158Q, D222*, R243Q, R243*, R252W, R261Q, R261*, E280K, P281L, A300S, I306V, S349P, IVS10-11G>A, E390G, A403V, R408Wp, Y414C, IVS12+1G>A. The total allele frequency for Russia is approximately 90% of all chromosomes in patients with variants in the PAH gene.”

Were the alleles of these 25 genetic variants in the PAH gene selected to be analyzed because they are highly prevalent in Russia? If so, why did you expect to find them in the population of a different ethnic origin?

In the M&M (page 10, lines 427-435), the authors described that they genotyped additional samples of 207 healthy unrelated individuals (of Ossetian ancestry up to the 3rd generation) to calculate their carriage of mutations in the PAH gene. Here again, the authors genotyped nine “frequent genetic variants for the Russian Federation and the North Caucasus region”, five of which were not found in the newborns in the North Ossetia-Alania in a 25-year period. So why did the authors expect to find any of these five variants, maybe typical for Russia but not found in newborns, in that additional Ossetian sample?

I highly, highly recommend having the manuscript read by a native English speaker.

I also do not understand why the authors did not write and use throughout manuscripts the rsID numbers of at least the two most common gene variants they found in the PAH gene anywhere in the manuscript, when that would increase paper`s visibility?

Why did you not write the sex of those 29 children who tested positive for PKU?

Abstract

The abstract is not clearly written. To begin with, for readers who might have not heard of these diseases, the information is missing that the HPA denotes a milder form of the disease characterized by abnormally elevated levels of phenylalanine, while classic PKU is a more severe form of the disease, and that both are the result of a mutation in the PAH gene. You later state that you found two variants in the PAH gene by analyzing samples of HPA patients, and then that a study of 29 patients of different ethnicities found PKU in 78% of the individuals - it is not clear that this is the same sample. You also wrote that the high frequency of the two variants in the Ossetian population is "possibly surpassing the global prevalence". You cannot use the term "possibly", you have to check the available data and conclude whether the prevalence is higher or not. You also wrote that your study “highlighted the relevance of considering maternal PKU syndrome in pregnancy planning due to its impact on fetal development”, but you did not investigate the role of maternal PKU syndrome on fetal development in your study.

Please write the abstract again based on your own results, by keeping it as simple as possible and taking into account the structure requested by the journal (headings are not needed, but there should be a background, methods, results, and conclusion).

Keywords - I think you need to include the term "PAH gene" and that the term "Ossetians" is ambiguous here - it is not clear whether you mean the inhabitants of the state or the ethnic origin of a part of the respondents.

page 2, lines 51-52 - when describing moderate and mild PKU, you must standardize the writing of the range: either "from X to Y mg/dl" or "X-Y mg/dl" should be written

page 2, lines 77-79 - here is a sentence in Cyrillic

page 3, Results

Table 1 – why is the 1st entry in the 1st column underlined?

I presume that the column entitled “Whole population”, refers to your total sample of newborns and not to that additional 207 people who were genotyped? If so, please be entitle it properly.

Table 2 – What does a letter “N” in the last two Ossetian samples mean? Why is the 1st entry under the title “Other ethnic groups” underlined? I suggest you use some other label for the two brothers from the same family because the asterisk sign already appears in the names of the genotypes in the second column.

page 9, Materials and Methods

line 376 – change into “The study included an analysis of the results of NBS for PKU …”

lines 382-383 “At the 1st stage sample collection in Maternity Hospitals were performed” should be changed into “In the 1st phase, samples were collected in maternity hospitals.”

line 392 – there is an extra “in”

Comments on the Quality of English Language

I highly recommend having the manuscript read by a native English speaker.

Author Response

First of all, we would like to thank the Reviewers for their in-depth and detailed analysis of our manuscript. We have tried to fulfill all their remarks and comments and believe the manuscript to be significantly improved.

Reviewer 2

The research describes the results of screening for two diseases, hyperphenylalaninemia (HPA) and phenylketonuria (PKU), in newborns from North Ossetia-Alania in a 25-year period. In the samples of newborns with positive screening results (immunofluorescence and measurement of phenylalanine concentration) whose parents agreed to further testing (n=29), the most frequent genetic variants in the PAH gene were searched using molecular genetic methods, and in case these previously described variants were not found, Sanger sequencing was performed. In total 28 patients, a combination of genotyping and sequencing revealed 18 previously noted variants in the PAH gene, while in one person there were found no changes in the PAH gene.

It seems to me that the genetic research on the blood samples of newborns was carried out methodologically correctly, however the manuscript has many shortcomings and inconsistencies. First of all, most of the manuscript is not written in the spirit of the English language, so it is difficult to follow. Not only that, but also often unexpected new moments occur.

Reply: Thank you for your feedback. We have conducted an additional review of the English language usage throughout the manuscript to ensure clarity and correctness.

I`ll show an example: the authors stated that the research was conducted on the population of the North Ossetia-Alania, which is in the Introduction described as an “Eastern Iranian ethnic group, communicating in the Ossetian Eastern Iranian language”. Then, in Table 2 in the Results section, ethnic background of parents of the PKU-affected children was written, and it was shown that although the most of the parents were of Ossetian origin, there were also some other ethnic groups.

Reply: We have incorporated demographic data on the Republic of North Ossetia-Alania into the Introduction section.

And finally, in the Materials and Methods (page 9, lines 399-401), after specifying which genetic variants in the PAH gene were analyzed, the authors wrote sentence that has no meaning and refers to allele frequency in Russia.

Reply: We have removed the sentence as it is not pertinent to the current study.

“1st stage includes analysis of 25 frequent genetic variants of the PAH gene included S16*, L48S, IVS2+5G>A, IVS2+5G>C, R111*, IVS4+5G>T, ex5del4154ins268, R158Q, D222*, R243Q, R243*, R252W, R261Q, R261*, E280K, P281L, A300S, I306V, S349P, IVS10-11G>A, E390G, A403V, R408Wp, Y414C, IVS12+1G>A. The total allele frequency for Russia is approximately 90% of all chromosomes in patients with variants in the PAH gene.” Were the alleles of these 25 genetic variants in the PAH gene selected to be analyzed because they are highly prevalent in Russia? If so, why did you expect to find them in the population of a different ethnic origin?

Reply: We excluded the mention of analyzing 90% of alleles as it is not relevant to the current study. The analysis of the aforementioned 25 frequent genetic variants in the PAH gene was conducted as the first stage of confirmatory DNA diagnosis. This test system covering 25 frequent variants was selected based on its proven high efficiency in confirming cases in our previous studies. Given the specific population under study, we did not anticipate it to detect a significant number of pathogenic alleles.

In the M&M (page 10, lines 427-435), the authors described that they genotyped additional samples of 207 healthy unrelated individuals (of Ossetian ancestry up to the 3rd generation) to calculate their carriage of mutations in the PAH gene. Here again, the authors genotyped nine “frequent genetic variants for the Russian Federation and the North Caucasus region”, five of which were not found in the newborns in the North Ossetia-Alania in a 25-year period. So why did the authors expect to find any of these five variants, maybe typical for Russia but not found in newborns, in that additional Ossetian sample?

Reply: We analyzed the carriage frequency of 9 variants in healthy individuals to compare it with the genotypes of patients identified through NBS. The detection of only P281L and P211T variants in the healthy population was consistent with the findings of the NBS.

I highly, highly recommend having the manuscript read by a native English speaker.

Reply: Thank you for your feedback. We have conducted an additional review of the English language usage throughout the manuscript to ensure clarity and accuracy.

I also do not understand why the authors did not write and use throughout manuscripts the rsID numbers of at least the two most common gene variants they found in the PAH gene anywhere in the manuscript, when that would increase paper`s visibility?

Reply: We have incorporated the rsIDs of the variants into Table 1 to enhance the completeness of the information provided.

Why did you not write the sex of those 29 children who tested positive for PKU?

Reply: We have included the sex of the patients as an additional column in Table 2, as per your request.

Abstract

The abstract is not clearly written. To begin with, for readers who might have not heard of these diseases, the information is missing that the HPA denotes a milder form of the disease characterized by abnormally elevated levels of phenylalanine, while classic PKU is a more severe form of the disease, and that both are the result of a mutation in the PAH gene. You later state that you found two variants in the PAH gene by analyzing samples of HPA patients, and then that a study of 29 patients of different ethnicities found PKU in 78% of the individuals - it is not clear that this is the same sample. You also wrote that the high frequency of the two variants in the Ossetian population is "possibly surpassing the global prevalence". You cannot use the term "possibly", you have to check the available data and conclude whether the prevalence is higher or not. You also wrote that your study “highlighted the relevance of considering maternal PKU syndrome in pregnancy planning due to its impact on fetal development”, but you did not investigate the role of maternal PKU syndrome on fetal development in your study.

Please write the abstract again based on your own results, by keeping it as simple as possible and taking into account the structure requested by the journal (headings are not needed, but there should be a background, methods, results, and conclusion).

Reply: Thank you for your guidance. We have revised the abstract to incorporate the valuable suggestions provided.

Keywords - I think you need to include the term "PAH gene" and that the term "Ossetians" is ambiguous here - it is not clear whether you mean the inhabitants of the state or the ethnic origin of a part of the respondents.

Reply: Thank you for your input. We have updated the keywords accordingly.

page 2, lines 51-52 - when describing moderate and mild PKU, you must standardize the writing of the range: either "from X to Y mg/dl" or "X-Y mg/dl" should be written

Reply: We have made the necessary corrections to the text.

page 2, lines 77-79 - here is a sentence in Cyrillic

Reply: Thank you for your thorough review. We have addressed the issue by replacing the sentence with an accurate one.

page 3, Results

Table 1 – why is the 1st entry in the 1st column underlined?

Reply: Thank you for bringing this to our attention. We have rectified the error by correcting the underlining of the 1st entry in the 1st column.

I presume that the column entitled “Whole population”, refers to your total sample of newborns and not to that additional 207 people who were genotyped? If so, please be entitle it properly.

Reply: Thank you for pointing out this issue. We have addressed it by correcting the title of the column.

Table 2 – What does a letter “N” in the last two Ossetian samples mean? Why is the 1st entry under the title “Other ethnic groups” underlined? I suggest you use some other label for the two brothers from the same family because the asterisk sign already appears in the names of the genotypes in the second column.

Reply: Thank you for bringing this to our attention. We have revised the content of Table 2 to enhance clarity.

page 9, Materials and Methods

line 376 – change into “The study included an analysis of the results of NBS for PKU …”

Reply: We have changed the text accordingly (now lines 395-396).

lines 382-383 “At the 1st stage sample collection in Maternity Hospitals were performed” should be changed into “In the 1st phase, samples were collected in maternity hospitals.”

Reply: We have changed the text accordingly (now lines 402-403).

line 392 – there is an extra “in”

Reply: We have changed the text accordingly (now line 414).

Round 2

Reviewer 2 Report (New Reviewer)

Comments and Suggestions for Authors

I recommend the manuscript in its current form for publication.

This manuscript is a resubmission of an earlier submission. The following is a list of the peer review reports and author responses from that submission.

Round 1

Reviewer 1 Report

Comments and Suggestions for Authors

The novel information in this manuscript gets lost between material that is included and material the reader might wish to have included.

For those who are unaware of the geography and ethnic relationships of the population, this material might be better introduced early in the manuscript.  Is the, or has the, Ossetian population been a genetic isolate so that variants found in PAH might be novel?

No information is given on the inferred 1 family with 2 affected individuals since 29 patients had confirmatory DNA analysis and they represented 28 unrelated families.

Is the sample of 29 individuals representative of the 46 identified hyperphenylalanemic infants (those who were detected on newborn screening)?

There is insufficient information about the two individuals in whom a single variant was found.  How deep into the intronic regions to urinalysis go?  What is the probability that there is an intronic splice site created that causes increased phenylalanine concentrations.  Furthermore it is not clear from what should be table 2 how the maximum phenylalanine concentration was detected nor at what age.

The reviewer is curious how you accessed the PAH database at McGill when that site has been unavailable for over a year to the reviewer.  Furthermore the more up-to-date database curated by Professor Blau would suggest that there are more mutations than implied in the manuscript.

How did you determine residual PAH activity ( line 153)? Was this taken from published information?

You need to comment on the difference between the calculated frequency of homozygous P281L, based on your sample 207 unrelated individuals, and the detected frequency.

Is the incidence of hyperphenylalaninemia implied on line 218 clinical incidence, genetic incidence or an incidence based on detectability on newborn screening?  The real incidence may be genetic and the detectable incidence of lower although you have identified at least 1 P211T homozygote.

There are other resources that would suggest your wording on line 197 is inaccurate.  ClinVar (www.ncbi.nlm.nih.gov/clinvar/) lists V177M unequivocally as a mild mutation and there is sufficient amount of data to say the same about R169H.  Given that no second variant was found for the individual with L83Wfs*9, he really cannot say whether this is mild or not.  It well-established, and you further support(lines 288-289) that the mild of the 2 mutations in an individual governs the phenotype.  If there is a mild but unrecognized variant on the other allele, the stop codon variant could be severe.

Reference 16 as cited on line 258 does not actually contain the variant and reference 25 is derived from a database.  The original information, from the database should be used in its place.

Reference 16 is also relevant for the first phrase of lines 263-265.

Reference 27 does not say anything about haplotypes and so is inappropriate for lines 268-270.  This information actually is found in Eisensmith RC et al 1995.

An improved set of references is needed for lines 82-87. Greeves et al (your reference 19) do not comment on whether the phenotype of an individual who is compound heterozygote for 2 "severe" mutations is milder than that of a homozygote for either variant. Kayaalp et al actually conclude that the HPA phenotype is more complex than that predicted by the mendelian inheritance of the alleles.

Did you do time resolved immunofluorescence or a fluorescence detection based enzyme assay to detect elevated phenylalanine concentrations in the dried blood spots?

More detail is needed about the detection of copy number variants especially for those readers not familiar with multiplex ligation-dependent probe amplification (MLPA). 

There is no information about longer-term follow-up and how intellectual disability was assessed, nor on the method for which postnatal blood phenylalanine concentrations were determined.

Where there are any individuals diagnosed symptomatically due to failure to be screened or false negative newborn screening?  This goes to your lines 403-404.  The hope of newborn screening is that no symptoms are present by the time of a confirmatory diagnosis.

Was any neuropsychological testing done on the individuals with milder hyperphenylalaninemia as specific cognitive function may be affected by even modest elevations in blood (and brain) phenylalanine?

How is the information in the paragraph beginning on line 413 novel?  The risk of maternal PKU should be included in your discussion because that is relevant to the clinical features of elevated phenylalanine concentrations in your population and the high frequency of individuals who are noncompliant with recommended therapy.

Comments on the Quality of English Language

On line 34, ‘attributed’ is probably the wrong word as the pathogenicity of the defect in phenylalanine-4-hydroxylase is not in doubt.

Similarly on line 50, "implicated" is the wrong word.  The entire phrase "implicated in the development of PKU" is really not needed.  The section from lines 52 through 54 does not add to this manuscript.

On line 81 what is meant by "ambiguous"?

The wording of line 253-254 should be clarified.  Codon refers to a DNA or RNA triplet rather than the amino acid.

There is a typographical error in line 39: FØlling (pronounced ‘eu’ and at least once in an authorship line of his work,  an umlaut was used to represent it as Fölling)

There are numerous instances where capitals are used and not needed.  Once the abbreviation PKU was introduced the word phenylketonuria become superfluous throughout the rest of the manuscript.  There are a number of instances of Phenylketonuria and Phenylalanine Hydroxylase.

Author Response

Q1.1: The novel information in this manuscript gets lost between material that is included and material the reader might wish to have included.

For those who are unaware of the geography and ethnic relationships of the population, this material might be better introduced early in the manuscript.  Is the, or has the, Ossetian population been a genetic isolate so that variants found in PAH might be novel?

A1.1: Thank you for the comment. We have extended the Introduction with information on the history of Ossetians as well as estimates of their genetic structure based on our previous studies (lines 90-107).

Q1.2: No information is given on the inferred 1 family with 2 affected individuals since 29 patients had confirmatory DNA analysis and they represented 28 unrelated families.

A1.2: Yes, one family has two brothers with PKU who did not strictly compliant with the diet therapy/ We specified this information in lines 132-134 as well as in Table 2 (labeled with an asterisk).

Q1.3: Is the sample of 29 individuals representative of the 46 identified hyperphenylalanemic infants (those who were detected on newborn screening)?

A1.3:  Thank you for the comment. Not all families were available for molecular genetic analysis since not all parents appeared interested in DNA diagnosis. We specified this in lines 130-131. We also have provided more details on the sample which underwent DNA-testing (lines 132-134).

Q1.4:There is insufficient information about the two individuals in whom a single variant was found.  How deep into the intronic regions to urinalysis go?  What is the probability that there is an intronic splice site created that causes increased phenylalanine concentrations.  Furthermore it is not clear from what should be table 2 how the maximum phenylalanine concentration was detected nor at what age.

A1.4: We could not exclude the existence of deep intronic or regulatory variants affecting the PAH locus since only analysis of coding exons of the PAH gene and +/- 20 bp adjacent intronic regions were searched for mutations. We specified this in the Materials and Methods section (lines 401). We also added the information that we could not exclude the presence of deep intronic or regulatory variants within the main text (lines 195-198). Phe level monitoring was performed every 3 to 6 months by the same way as in NBS. The maximum value was recorded for the entire follow-up period for each patient, but the exact age was not recorded. This information is added to the M&M section (lines 417-420).

Q1.5:The reviewer is curious how you accessed the PAH database at McGill when that site has been unavailable for over a year to the reviewer.  Furthermore the more up-to-date database curated by Professor Blau would suggest that there are more mutations than implied in the manuscript.

A1.5: Thank you for the comment. Indeed, the PAH database at McGill is currently unavailable, this information remains from the previous publications of our laboratory. We have duly noted the update regarding the usage of the biopku.org resource.

Q1.6: How did you determine residual PAH activity ( line 153)? Was this taken from published information?

A1.6: The residual PAH activity was taken from the published information, firstly, from the Himmelreich et al. (2018) manuscript. Other sources are also mentioned in the Notes to the Table 1.

Q1.7: You need to comment on the difference between the calculated frequency of homozygous P281L, based on your sample 207 unrelated individuals, and the detected frequency.

A1.7: We have updated the section on the estimated and observed allelic frequencies (lines 267-274).

Q1.8: Is the incidence of hyperphenylalaninemia implied on line 218 clinical incidence, genetic incidence or an incidence based on detectability on newborn screening?  The real incidence may be genetic and the detectable incidence of lower although you have identified at least 1 P211T homozygote.

A1.8: line 218 (now 262) contains information about observed through NBS incidence of HPA (1:4864). We expanded the comparison of observed vs estimated values of incidence within this paragraph (lines 254-261).

Q1.9: There are other resources that would suggest your wording on line 197 is inaccurate.  ClinVar (www.ncbi.nlm.nih.gov/clinvar/) lists V177M unequivocally as a mild mutation and there is sufficient amount of data to say the same about R169H.  Given that no second variant was found for the individual with L83Wfs*9, he really cannot say whether this is mild or not.  It well-established, and you further support(lines 288-289) that the mild of the 2 mutations in an individual governs the phenotype.  If there is a mild but unrecognized variant on the other allele, the stop codon variant could be severe.

A1.9: Thank you for bringing this to our attention. We acknowledge the oversight in the classification of the L83Wfs*9 variant in our manuscript. Upon reevaluation, we recognize that there was insufficient information available to support its classification as a mild mutation. We have accordingly revised our discussion of this variant to accurately reflect its potential severity. We appreciate your diligence in reviewing our work and ensuring the accuracy of our findings. If you have any further questions or concerns, please do not hesitate to let us know.

Q1.10: Reference 16 as cited on line 258 does not actually contain the variant and reference 25 is derived from a database.  The original information, from the database should be used in its place.

A1.10: Thank you for your attention. We have corrected the references accordingly.

Q1.11: Reference 16 is also relevant for the first phrase of lines 263-265.

A1.11: Thank you for your attention. We have corrected the references accordingly.

Q1.12: Reference 27 does not say anything about haplotypes and so is inappropriate for lines 268-270.  This information actually is found in Eisensmith RC et al 1995.

A1.12: Thank you for your attention. We have duly noted your comment and removed the sentence from the manuscript as it was not relevant to the current study.

Q1.13: An improved set of references is needed for lines 82-87. Greeves et al (your reference 19) do not comment on whether the phenotype of an individual who is compound heterozygote for 2 "severe" mutations is milder than that of a homozygote for either variant. Kayaalp et al actually conclude that the HPA phenotype is more complex than that predicted by the mendelian inheritance of the alleles.

A1.13: Thank you for your attention. We have duly noted your comment and removed this controversial assertion from the manuscript as it was not relevant to the current study.

Q1.14: Did you do time resolved immunofluorescence or a fluorescence detection based enzyme assay to detect elevated phenylalanine concentrations in the dried blood spots?

A1.14: We specified the method for NBS in the Materials & Methods section. We used time resolved immunofluorescence (lines 381-383).

Q1.15: More detail is needed about the detection of copy number variants especially for those readers not familiar with multiplex ligation-dependent probe amplification (MLPA).

A1.15: We described the MLPA method in the Materials & Methods section.

Q1.16: There is no information about longer-term follow-up and how intellectual disability was assessed, nor on the method for which postnatal blood phenylalanine concentrations were determined.

A1.16: We specified the follow-up as well as methodology of Phe level monitoring in the Materials & Methods section.

Q1.17: Where there are any individuals diagnosed symptomatically due to failure to be screened or false negative newborn screening?  This goes to your lines 403-404.  The hope of newborn screening is that no symptoms are present by the time of a confirmatory diagnosis.

A1.17: There were no referrals of patients with suspected PKU outside of NBS during the study.

Q1.18: Was any neuropsychological testing done on the individuals with milder hyperphenylalaninemia as specific cognitive function may be affected by even modest elevations in blood (and brain) phenylalanine?

A1.18: Neuropsychological developmental assessments of patients with mHPA have not been conducted and are not anticipated in the Russian Federation. Currently, evaluations of individuals with PKU primarily focus on determining the presence or absence of intellectual impairment for disability determination purposes. Comprehensive neuropsychological assessments for patients with mHPA are not part of routine clinical practice in this context.

Q1.19: How is the information in the paragraph beginning on line 413 novel?  The risk of maternal PKU should be included in your discussion because that is relevant to the clinical features of elevated phenylalanine concentrations in your population and the high frequency of individuals who are noncompliant with recommended therapy.

A1.19: Thank you for your valuable comment. We moved this paragraph to the Discussion section.  

Q1.20: On line 34, ‘attributed’ is probably the wrong word as the pathogenicity of the defect in phenylalanine-4-hydroxylase is not in doubt.

A1.20: We have changed the text accordingly.

Q1.21: Similarly on line 50, "implicated" is the wrong word.  The entire phrase "implicated in the development of PKU" is really not needed.  The section from lines 52 through 54 does not add to this manuscript.

A1.21: We have changed the text accordingly.

Q1.22: On line 81 what is meant by "ambiguous"?

A1.22: We have changed the text accordingly. We wanted to highlight that different mutations possesses different residual functional activity of PAH.

Q1.23: The wording of line 253-254 should be clarified.  Codon refers to a DNA or RNA triplet rather than the amino acid.

A1.23: We have changed the text accordingly.

Q1.24: There is a typographical error in line 39: FØlling (pronounced ‘eu’ and at least once in an authorship line of his work,  an umlaut was used to represent it as Fölling)

A1.24: Thank you for your attention. We have corrected the surname of Dr. Asbjørn Følling within the text as well as in the ref. [2].

Q1.25: There are numerous instances where capitals are used and not needed.  Once the abbreviation PKU was introduced the word phenylketonuria become superfluous throughout the rest of the manuscript.  There are a number of instances of Phenylketonuria and Phenylalanine Hydroxylase.

A1.25: We changed the text accordingly trying to use abbreviations more consistently.

Reviewer 2 Report

Comments and Suggestions for Authors

This study contributes analysis of distribution of pathogenic genomic variants associated with hyperphenylalaninemia (HPA) and phenylketonuria (PKU) in the Republic of North Ossetia-Alania during period of 1997-2022. This is important contribution to population and clinical genetics. Paper presents well written clear descriptive analysis of distribution of PKU associated genetic variants in a small RNO-Alania population collected by newborn screening program.   

A few comments below. 

First comment regards how variant describing  nomenclature is used. Mostly the variants of PAH gene are described, however in a few places a PTS gene is also mentioned.  It is confusing where in some instances a full HGVSc variant identifier is given (  chr position ) ; in other instances only a substitution in a coding part ( transcript is assumed) .   Although in methods you mention that HGVS nomenclature is used to denote variants, it is not consistent everywhere. I think the notations must be consistent.  It would be very helpful if all mutations identified by a protein change throughout the article ( Table 1 ) would have a full HGVS notation  regardless  of  a repetition of the transcript identifier. In general it will improve readability and will make it easier to query variant databases if necessary. The PTS variants could be summarised in a similar way.    

I am a bit confused with the expressions "variant detected on XX chromosomes" - lines 128, 133, 137, 142 and few other instances. Please explain what do you mean by that. It literally gives an impression that PAH gene variant was detected on  a number of different chromosomes, and then it rises a question "what does it mean variant detected on multiple chromosomes?" if the variant is unique and is located on a definite transcript and position? 

Line 357 please explain exactly how you applied a fomula:  2pq=m/N. In that formula which variables were given and which were calculated? Were the m and N  given and the 2pq was calculated or it is vice versa?  This is not clear from the text. Also, please define what p and q means if they need to be shown in the formula.     

Line 111 define RCMG abbreviation as in line 343, since  the abbreviation RCMG appears first time on  line 111 in the text.

Comments on the Quality of English Language

Overall it is clear, except of some semantics of the expressions , please see the comments above. 

Author Response

This study contributes analysis of distribution of pathogenic genomic variants associated with hyperphenylalaninemia (HPA) and phenylketonuria (PKU) in the Republic of North Ossetia-Alania during period of 1997-2022. This is important contribution to population and clinical genetics. Paper presents well written clear descriptive analysis of distribution of PKU associated genetic variants in a small RNO-Alania population collected by newborn screening program.  

A few comments below.

Q2.1: First comment regards how variant describing  nomenclature is used. Mostly the variants of PAH gene are described, however in a few places a PTS gene is also mentioned.  It is confusing where in some instances a full HGVSc variant identifier is given (  chr position ) ; in other instances only a substitution in a coding part ( transcript is assumed) .   Although in methods you mention that HGVS nomenclature is used to denote variants, it is not consistent everywhere. I think the notations must be consistent.  It would be very helpful if all mutations identified by a protein change throughout the article ( Table 1 ) would have a full HGVS notation  regardless  of  a repetition of the transcript identifier. In general it will improve readability and will make it easier to query variant databases if necessary. The PTS variants could be summarised in a similar way.   

A2.1: Thank you for your attention. We changed the naming of the variants consistently throughout the manuscript.

Q2.2: I am a bit confused with the expressions "variant detected on XX chromosomes" - lines 128, 133, 137, 142 and few other instances. Please explain what do you mean by that. It literally gives an impression that PAH gene variant was detected on  a number of different chromosomes, and then it rises a question "what does it mean variant detected on multiple chromosomes?" if the variant is unique and is located on a definite transcript and position?

A2.2: In that context, "variant detected on XX chromosomes" means that allele count of the particular allele is XX alleles. We changed this phrase throughout the manuscript accordingly in order to avoid misunderstanding.

Q2.3: Line 357 please explain exactly how you applied a fomula:  2pq=m/N. In that formula which variables were given and which were calculated? Were the m and N  given and the 2pq was calculated or it is vice versa?  This is not clear from the text. Also, please define what p and q means if they need to be shown in the formula.    

A2.3: We updated the information about the estimation of the heterozygous carriage rate in the M&M section. (lines 432-435).

Q2.4: Line 111 define RCMG abbreviation as in line 343, since  the abbreviation RCMG appears first time on  line 111 in the text.

A2.4: We changed the text accordingly.

Round 2

Reviewer 1 Report

Comments and Suggestions for Authors

How is the history of mutational assessment, lines 51 through 58 relevant to the rest of the paper?  Lines 59 and 60 are relevant but need to have an anchoring comment that could be lines 67 through 70.  The

The comments on lines 62 through 66 best goes with the paragraph beginning on line 77.  A reordering would improve clarity 

The addition of the paragraph beginning on line 88 has made a significant improvement for the reader.

In table 2, disability versus intellectual disability is unclear.  This needs to be clarified within the methods - materials section.

In table 1 you list the V177M variant as having undetermined residual activity but in the newly inserted results section, a reader could infer that the literature has such data rather than your statement relying on historic blood phenylalanine concentrations from a variety of individuals with different ethnicities.  Similar potential issue with the R169H variant.  Also, Pey et al used computer modeling to predict pathogenic classification.

Lines 225-228' is discussion, not results.

Please explain why the confidence interval on line 236 and that on line 243 do not match.

Portion of lines 244 through 258 should be in the discussion rather than in the results as this is derived from your data and you expound on the significance of that derivation.

The statement leading line 323 requires a reference.  Then the sentence concluding on line 325 could be made specific to your studied population.

Why was a retrospective evaluation of medical records for all patients diagnosed with PKU before the implementation of newborn screening conducted when that information is not included within this manuscript?  That statement does not seem relevant to the methods and materials.  If it was relevant what data was utilized?

Since you did not actually determine protein activity you cannot use the word 'determined' on line 164.  You 'assigned' the status based on information in the literature.

The term 'dynamics' needs to be clarified.  If you meant 'in clinic evaluations or in monitoring situations', a re-working probably would help.

In the sentence beginning on line 204, the phrase strictly adhering to dietary therapy, is probably not needed.

On line 366, was a second sample obtained for confirmation of the newborn screening results?  If so, the symptoms need to be rewritten to more clearly identify this.

The last line of the conclusion is not supported by any data that you provide but would be relevant for discussion section.  The theme of the lines 453-458 should also be in the discussion.

If no individual was subsequently diagnosed with PKU if they had undergone newborn screening, this should be included in your results section.  It would also obviate the need for the second half of the sentence on line 449.

Overall this reviewer had the impression that there are statements in the results and conclusions sections which are better placed within the discussion so that your conclusions become a shorter section that would tend to stand out more to the reader.   The conclusion section is made more confusing because it is not clear in some cases when you are talking about the variant frequency in the newborn screening population and the variant frequency in the 207 unrelated individuals.

Comments on the Quality of English Language

There are number of instances where the English language/utilization needs to be improved.

These include lines 45-46 with the word averages used twice in 1 sentence.

On lines 155-156, changing wording would improve the ability of the sentence..  Perhaps something like 1 instance of blank number of various other variants with different residual activities were found in the heterozygous state.

In table to there is a phrase in Cyrillic which either needs to be translated or removed.  You implies noncompliance to diet recommendations, elsewhere in the text.

On line 215 a space is needed between R169H and ‘are’

The expanded description of the genetic variant in table 1 needs a little bit of respacing so that the 2 words legacy name are clearly separated.

The symptoms starting line 284 needs to be reworked since the phrase beginning ‘comparable to’ seems to refer back to the frequency in Russia rather than the frequency in the Ossetian population.

On line 302 consider a change from ‘major’ to ‘represents the majority allele’

There are still instances where phenylalanine, newborn screening or phenylalanine hydroxylase is capitalized and does not need to be capitalized.

On lines 390-391, you have already introduced MGC and RCCH so that the terms do not need to be written out